# Boron Hydrogen Compounds: Hydrogen Storage and Battery Applications

**DOI:** 10.3390/molecules26247425

**Published:** 2021-12-07

**Authors:** Hans Hagemann

**Affiliations:** Département de Chimie Physique, Université de Genève, 30, Quai E. Ansermet, CH1211 Geneva 4, Switzerland; hans-rudolf.hagemann@unige.ch

**Keywords:** boron hydrides, hydrogen storage, solid ionic conductors

## Abstract

About 25 years ago, Bogdanovic and Schwickardi (B. Bogdanovic, M. Schwickardi: *J. Alloys Compd.* 1–9, 253 (1997) discovered the catalyzed release of hydrogen from NaAlH_4_. This discovery stimulated a vast research effort on light hydrides as hydrogen storage materials, in particular boron hydrogen compounds. Mg(BH_4_)_2_, with a hydrogen content of 14.9 wt %, has been extensively studied, and recent results shed new light on intermediate species formed during dehydrogenation. The chemistry of B_3_H_8_^−^, which is an important intermediate between BH_4_^−^ and B_12_H_12_^2−^, is presented in detail. The discovery of high ionic conductivity in the high-temperature phases of LiBH_4_ and Na_2_B_12_H_12_ opened a new research direction. The high chemical and electrochemical stability of *closo*-hydroborates has stimulated new research for their applications in batteries. Very recently, an all-solid-state 4 V Na battery prototype using a Na_4_(CB_11_H_12_)_2_(B_12_H_12_) solid electrolyte has been demonstrated. In this review, we present the current knowledge of possible reaction pathways involved in the successive hydrogen release reactions from BH_4_^−^ to B_12_H_12_^2−^, and a discussion of relevant necessary properties for high-ionic-conduction materials.

## 1. Introduction

Boron hydrogen compounds have been intensively studied for almost a century since the pioneering studies of A. Stock [1]. Boron hydrogen compounds are also energetic materials and were considered as rocket or jet fuels [2]; however, the toxicity of boranes has prevented their extended use. Currently, nontoxic compounds such as ammonia-borane are also studied as hypergolic propellants [3,4]. Recently, many different applications of boron hydrogen compounds have emerged [5]. In particular, compounds derived from *closo*-hydroborates such as B_12_H_12_^2−^ have found many new applications, including new all-solid-state batteries, medical applications, and as catalysts [6,7,8,9,10,11]. Since the discovery of catalyzed hydrogen release in NaAlH_4_ by Bogdanovic and Schwickardi [12], light boron and aluminum hydrides were intensively studied and reviewed as potential hydrogen storage materials [13,14,15,16,17,18,19,20,21,22]. The dehydrogenation reactions of metal borohydrides ultimately lead to hydrogen, metal and boron, or metal borides. In this reaction process, intermediate species are formed, particularly compounds with *closo*-hydroborate anion B_12_H_12_^2−^ [23,24]. B_12_H_12_^2−^ is particularly stable and can thus also act as a detrimental thermodynamic sink for further dehydrogenation reactions. The properties of *closo*-hydroborates and related anions were addressed in several recent publications [6,25,26,27,28]. New research on the thermal properties of *closo*-hydroborate salts revealed a high-temperature phase transition in Na_2_B_12_H_12_ leading to a superionic phase [29]. Thus, the controlled dehydrogenation of a borohydride salt can be used to safely prepare new *closo-* and *nido-* hydroborate salts for potential battery applications [30] without using toxic boranes such as B_10_H_14_, which were used for the synthesis of this large boron species [31].

In this review, we first describe experimental results on hydrogen storage in Mg(BH_4_)_2_, which has a large hydrogen content of 14.9 wt %. Hydrogen storage in other borohydrides, such as LiBH_4_, was recently reviewed [32]. Recent results on potential dehydrogenation intermediates for Mg(BH_4_)_2_ provide new insights on the potential reaction intermediates and are reported here. In this context, we then present recent results based on DFT calculations to explore possible reaction paths for successive dehydrogenation reactions starting from BH_4_^−^. These paths are described in more detail in the following section, which discusses the formation and reactions of B_3_H_8_^−^, as this ion is considered to be one of the reaction intermediates during the dehydrogenation of borohydride compounds. The high-temperature dehydrogenation of B_3_H_8_^−^ leads to the formation of *closo*-hydroborate anions B_10_H_10_^2−^ and B_12_H_12_^2−^, which form excellent solid ionic conductors for new all-solid-state batteries [30]. The properties of these ionic conductors are presented in the last section.

## 2. Magnesium Borohydride

Among the many compounds considered for hydrogen storage, Mg(BH_4_)_2_ is particularly interesting and has been studied by many authors. The earlier studies on Mg(BH_4_)_2_ were reviewed in detail in 2016 [22]. Mg(BH_4_)_2_ has a hydrogen content of 14.9 mass % [22,33]. This compound can be prepared in different crystalline modifications, and high pressure-phase transitions were also observed [33]. Porous γ-Mg(BH_4_)_2_ can also adsorb 0.8 H_2_ at low temperatures and 100 bar to achieve a total hydrogen mass content of 17.4% [33]. High-pressure phase δ-Mg(BH_4_)_2_ has a very high volumetric hydrogen content of 147 g H_2_/L. Mg(BH_4_)_2_ can also form amorphous solids. Overall dehydrogenation reaction
Mg(BH_4_)_2_ ➔ MgB_2_ + 4 H_2_
is, in fact, a multistep reaction (see Figure 1) with various reaction intermediates, such as Mg(B_3_H_8_)_2_, MgH_2_, and MgB_12_H_12_, which were proposed both experimentally and theoretically [22,34,35,36]. MgB_2_ is the decomposition product obtained after heating to 500 °C [37]. Boron-rich MgB_7_ films are obtained by heating volatile Mg(B_3_H_8_)_2_ solvates with dimethyl ether and diethyl ether [38].

MgB_2_ can be rehydrogenated, although under drastic conditions (950 bar H_2_ at 400 °C) [40]. The rehydrogenation of MgB_2_ can be accelerated with THF, MgH_2_, and Mg [41]. Mechanically milled mixtures of MgB_2_, THF, and 40 mol % Mg could thus absorb 6 wt % of H_2_ at 300 °C under 700 bar of H_2_, which is less drastic than that without THF. Recently, rehydrogenation at room temperature with mechanical activation by ball milling was reported [42]. These rehydrogenation reactions of MgB_2_ demonstrate the principle that hydrogen storage in Mg(BH_4_)_2_ is indeed reversible. A recent combined experimental and theoretical study concluded that the initial stages of rehydrogenation are associated with the formation of σ bonds of hydrogen with boron on the reactive edges of the MgB_2_ solid [43]. The rehydrogenation of intermediate compounds was also studied. MgB_3_H_8_.THF can be rehydrogenated under milder conditions than those of dry MgB_3_H_8_ (50 bar H_2_ and 200 °C for 5 h vs. 120 bar H_2_ and 250 °C for 48 h) [44]. MgH_2_ is formed in intermediate reaction steps, such as
6 Mg(BH_4_)_2_ ➔ MgB_12_H_12_ + 5 MgH_2_ + 13 H_2_

Magnesium hydride dissociates into Mg and H_2_ at high temperatures and low H_2_ pressures. The different reaction products observed under various conditions (see Figure 1) show that the reaction kinetics can be influenced by various parameters, which also include the initial crystalline modification of Mg(BH_4_)_2_.

The overall enthalpy of reaction for the dehydrogenation of Mg(BH_4_)_2_ (Δ_f_H° = −208 kJ/mol) to form MgB_2_ (Δ_f_H° = −91.96 kJ/mol) and hydrogen can be calculated [45,46,47] to be equal to +116 kJ/mol, i.e., less than 30 kJ/mol per hydrogen molecule released, which is, in principle, in the correct range for a hydrogen storage material [13]. The first step of a dehydrogenation reaction of BH_4_^−^ is likely to be the breaking of a B–H bond. Isotope exchange reactions of Mg(BH_4_)_2_ with D_2_ allow for producing a complete exchange to form Mg(BD_4_)_2_, and the corresponding activation energy was estimated to be about 51 kJ/mol [48]. For Ca(BH_4_)_2_, the corresponding activation energy was found to be 82 and 98.5 kJ/mol for the reverse reaction, confirming that breaking a bond with hydrogen or deuterium is the rate-limiting step [49]. Theoretical calculations of potential defects in Mg(BH_4_)_2_ suggest that, in the initial phase of the dehydrogenation, a H^−^ ion is formed that can diffuse in the lattice [50]. On the other hand, gas diffusion in the solid is also a contribution to exchange kinetics, as was shown by isotope exchange reactions with the highly porous modification of γ-Mg(BH_4_)_2_ with a high surface area compared to a ball-milled sample with a strongly reduced surface aera [51].

The reaction kinetics of hydrogen release in Mg(BH_4_)_2_ can be significantly enhanced by various additives, such as TiCl_3_ [52] or NbF_5_ and TiO_2_ [53]. Lewis bases in the form of solvates of Mg(BH_4_)_2_ can also accelerate the hydrogen release [54]. As shown in Figure 1, the THF solvate releases H_2_ gas below 200 °C to form Mg(B_10_H_10_). The formation of B_3_H_8_^−^ and B_12_H_12_^2−^ was also observed, but with THF and dimethyl ether, B_12_H_12_^2−^ remained a minor reaction product. The physical properties of Mg(BH_4_)_2_.3THF were recently investigated in detail [55]. In this compound, Mg^2+^ is coordinated to 2 BH_4_^−^ ions and 3 THF molecules. The orientational mobilities of the BH_4_^−^ ions are not particularly sensitive to the presence of THF. The authors concluded that “the presence of THF also disrupts the stability of the crystalline phase leading to enhanced kinetics for the dehydrogenations”. Recently, Tran et al. [56] reported that the presence of different glymes with Mg(BH_4_)_2_ results in various ratios of MgB_10_H_10_ and MgB_12_H_12_ upon thermolysis at 160–200 °C, and allows for selectively obtaining MgB_10_H_10_ with one equivalent of monoglyme. Mixtures of Mg(BH_4_)_2_ with (CH_3_)_4_NBH_4_ (5:1 molar) reveal reversible melting around 180–195 °C [57] with enhanced stability compared to melts of pure Mg(BH_4_)_2_ and (CH_3_)_4_NBH_4_. [Ph_4_P]_2_[Mg(BH_4_)_4_] gradually loses mass over 225–230 °C, but heating to 500 °C does not lead to the mass loss expected for the formation of MgB_2_. A similar behavior was observed for [Me_4_N]_2_[Mg(BH_4_)_4_] [58]. These findings suggest that derivates of Mg(BH_4_)_2_ with organic cations are rather stabilized.

Solvent-free Mg(B_3_H_8_)_2_ can be prepared by ball milling MgBr_2_ with NaB_3_H_8_ [38,59]. Kim et al. [38] reported the formation of boron-rich MgB_7_ films upon heating under vacuum above 425 °C due to some evaporation of Mg under these conditions. Thermogravimetry (TG) experiments [59] revealed a 30 wt % mass loss setting in above ca 80 °C corresponding to the evolution of B_2_H_6_, B_5_H_9_ and H_2_. The residual solid after heating to 200 °C was a mixture of mainly Mg(BH_4_)_2_, Mg(B_10_H_10_), and Mg(B_12_H_12_), and the combined evolution of H_2_, B_2_H_6_, and B_5_H_9_ was confirmed by mass spectrometry [60]. The addition of activated (ball-milled) MgH_2_ to Mg(B_3_H_8_)_2_ results in a strong reduction in borane evolution and up to 88% conversion back to Mg(BH_4_)_2_ at 100 °C. The presence of activated MgH_2_ thus substantially decreases the formation of (*closo*-hydro)borates and provides the necessary hydrogen for the conversion of B_3_H_8_^−^ back into BH_4_^−^.

These experiments suggest that, while Lewis acids may favor the dehydrogenation reactions of Mg(BH_4_)_2_, they do not necessarily catalyze the rehydrogenation reactions, as transition metal halides do not appear to affect the rehydrogenation of MgB_2_ [40,61]. THF and other Lewis bases appear to accelerate both the dehydrogenation and rehydrogenation reactions of Mg(BH_4_)_2_, and encourage more studies to even further improve the kinetics.

## 3. DFT Calculations

The results presented above for Mg(BH_4_)_2_ suggest the formation of various intermediate species such as B_2_H_6_^2−^, B_3_H_8_^−^, B_4_H_10_^2−^, B_5_H_9_^2−^ and the *closo*-borates B_n_H_n_^2−^ (*n* = 8–12). For hydrogen storage applications, the only gaseous species resulting from dehydrogenation reactions should be hydrogen; thus, neutral boranes are a priori not involved in the reaction mechanisms. Many other anionic boron hydrides have been reported in the literature and could be involved in one reaction step or another. In 1999, some reactions between neutral and anionic boron hydrides related to the formation of B_3_H_8_^−^, B_5_ anions, and some other species were reviewed [62].

In order to assess the driving forces for different reactions, thermodynamic information can be very useful, but experimental data are very scarce. For alkali borohydrides, thermodynamical data are available [47], but only few other experimental data are available. Using the experimental values of the formation enthalpy of Mg(BH_4_)_2_ [45] and La(BH_4_)_3_ [63], the formation enthalpy of other M(BH_4_)_2_ and M(BH_4_)_3_ compounds were estimated, assuming that the lattice enthalpy of bromides and borohydrides with the same metal ion were identical within about 15 kJ/mol [46]. The experimental formation enthalpy of NH_4_B_3_H_8_ (−530 ± 33 kJ/mol) [64], (NH_4_)_2_B_10_H_10_ (−359.2 ± 10 kJ/mol) [65], and of guanidinium and other nitrogen-based *closo*-borates was reported [66]. Recently, new heat capacity measurements for Na, K, Rb, Cs, Mg, Ca borohydrides were reported [67]. The knowledge of all thermodynamic properties in principle allows for quantitatively describing the phase diagram of a system, which was performed using available data for the Mg–B–H system [68].

In the absence of experimental data, theoretical data are obtained. It is quite challenging to obtain accurate results of formation enthalpies using DFT. Nguyen et al. [69] calculated for the formation enthalpy of (NH_4_)_2_B_10_H_10_ with the G3 method the value of −184 kJ/mol, which is quite different from the experimental value of −359.2 kJ/mol. For α-Mg(BH_4_)_2_, formation enthalpy values ranging from −67 to −277 kJ/mol were reported in the literature [68], while the experimental value was −208 kJ/mol [45]. Zhang et al. [23] computed relative formation energies of potential solid intermediates formed during the dehydrogenation of Mg(BH_4_)_2_, in combination with a Monte Carlo-based structure prediction method. They predicted a potential Mg_3_(B_3_H_6_)_2_ intermediate with a B_3_H_6_^3−^ ion, while Mg(B_3_H_8_)_2_ was found to be very high in relative energy and thereby unlikely to be formed.

The principal difficulty for estimating the formation enthalpy of crystalline solids is the evaluation of lattice energy, as different approaches (volume-based, Kaputinski equation etc.) lead to different values. Further, lattice energies can only be computed for crystalline materials, preferentially on the basis of experimental structure data, but experiments showed that a significant fraction of the reaction intermediates remain amorphous, complicating things even further.

DFT calculations in the gas phase are quite reliable, and allow for obtaining good structural data and vibrational frequencies, in particular when anharmonicity is included. Several studies report the formation enthalpy of borohydride ions in the gas phase [69,70,71,72] Anharmonic DFT calculations allow for obtaining improved agreement with experimental vibrational spectra, from which heat capacity data were calculated [73]. Figure 2 compares experimental [74] and DFT calculated [69,70,71,72] formation enthalpy data for neutral and anionic boron hydrogen species. Figure 2 shows that the calculated formation enthalpy for a given species (e.g., B_3_H_8_^−^) can differ by about 100 kJ/mole for different sources. These values are derived, for instance, from isodesmic reactions with known formation heat [69], thus generating a potential propagation of errors if the initial formation enthalpy values are different. We outline all reported values to highlight the limitations of the accuracy of these data.

Figure 2 shows that the experimental formation enthalpies of neutral species are all positive [74], with values ranging from 36 kJ/mol (for B_2_H_6_) to 210 kJ/mol for (B_2_H_4_). Gas phase reaction
2 B_2_H_6_ ➔ B_4_H_10_ + H_2_
has an enthalpy change of 66.1 − 2 × 36.4 = −6.7 kJ/mol, and shows that increasing the number of boron atoms in the cluster can be thermodynamically favorable for neutral species. Other reactions towards larger hydroboranes may become favorable at higher temperatures from the liberation of hydrogen. The first theoretical studies of enthalpy changes for reactions of neutral boranes were reported by M.L. McKee in 1990 [70], who showed that a sequence of BH_3_ additions followed by H_2_ elimination from B_2_H_6_ to B_6_H_10_ is overall exothermic, but with two less stable reaction intermediates (B_3_H_9_ and B_4_H_8_) that can act as barrier steps for the kinetics. Figure 2 shows that anionic species with 9–12 boron atoms are the most stable, which indicates that there is a thermodynamic driving force towards these anions. The most stable species in this figure is the *closo* B_12_H_12_^2−^ ion, and its stability is related to its 3-dimensional aromaticity [6]. The formation enthalpy of B_12_H_12_^2−^ in the gas phase was estimated to be between −325.5 and −428.6 kJ/mol according to different theoretical studies [72,75,76]. One key intermediate in the overall dehydrogenation reactions of BH_4_^−^ appears to be ion B_3_H_8_^−^, which is discussed in the next section.

## 4. Formation and Reactions of B_3_H_8_^−^

As mentioned above, the formation of Mg(B_3_H_8_)_2_ was observed during the decomposition of Mg(BH_4_)_2_ under dynamic vacuum [54,77], and Y(B_3_H_8_)_3_ was obtained after heating Y(BH_4_)_3_ under hydrogen pressure of 1–10 bar [78]. There are several reports in the literature on the synthesis of B_3_H_8_^−^ that highlight that various routes can lead to this ion. Starting from diborane under strongly reducing conditions, dianion B_2_H_6_^2−^ was reported to form [62,79]
2 B_2_H_6_ + 2 C_8_H_10_^−^ ➔ [BH_3_^2−^] + BH_3_ + 2 C_8_H_10_ ➔ [B_2_H_6_^2−^] + 2 C_8_H_10_

BH_3_^2−^ and B_2_H_6_^2−^ intermediates were identified by NMR. The reaction of B_2_H_6_^2−^ with additional diborane yields B_3_H_8_^−^ + BH_4_^−^, and no further intermediate was observed:B_2_H_6_ + B_2_H_6_^2−^ ➔ B_3_H_8_^−^ + BH_4_^−^

Another reaction observed was the reaction of potassium metal with THF.BH_3_ [80].
2 K + 4 THF.BH_3_ ➔ 2 K^+^ + B_3_H_8_^−^ + BH_4_^−^

Beall and Gaines [62] argue that also in this case, B_2_H_6_^2−^ is the reaction intermediate, which can then react with THF–BH_3_ to form either B_2_H_5_^−^ + BH_4_^−^ with the addition of the 4th THF.BH_3_ B_3_H_8_^−^ or first with THF–BH_3_ the ion B_3_H_9_^2−^, which then reacts with THF.BH_3_ to yield B_3_H_8_^−^ + BH_4_^−^. B_3_H_8_^−^ can also be formed from the reaction of BH_4_^−^ with diborane [81]:BH_4_^−^ + B_2_H_6_ ➔ B_3_H_8_^−^ + H_2_
BH_4_^−^ + B_2_H_6_ ➔ BH_3_ + B_2_H_7_^−^ ➔ BH_3_ + B_2_H_5_^−^ + H_2_ ➔ B_3_H_8_^−^ + H_2_
BH_4_^−^ + B_2_H_6_ ➔ BH_3_ + B_2_H_7_^−^ ➔ B_3_H_10_^−^ ➔ B_3_H_8_^−^ + H_2_

This reaction can proceed either via B_2_H_7_^−^ (hydride transfer) and B_3_H_10_^−^ (BH_3_ addition) followed by H_2_ detachment or via B_2_H_7_^−^, which first loses H_2_ to form B_2_H_5_^−^, which then adds BH_3_. The efficient synthesis of alkali metal octahydrotriborates (M = Na, K, Rb, Cs) from the reaction of MBH_4_ with 2 equivalents of dimethyl sulfide borane was reported [82]. The formation of ion B_2_H_7_^−^ was observed by NMR for the reaction of LiBH_4_ with THF.BH_3_ in THF [83], and during the solvothermal reaction of BH_4_^−^ with CH_2_Cl_2_ at 70 °C [84]. The reaction of BD_4_^−^ requires higher temperatures (90 °C) [84], which suggests that the rate-determining reaction step is associated with the breaking of a boron–hydrogen (deuterium) bond, which could be the formation of a reactive Lewis adduct of BH_3_ from BH_4_^−^, which then reacts with other BH_4_^−^ to form B_2_H_7_^−^ etc., as outlined above.

Once formed, B_3_H_8_^−^ can further react to yield B_9_ to B_12_ hydroborate anions. Using the DFT calculation formation enthalpies of B_9_H_14_^−^, B_3_H_8_^−^ and BH_4_^−^ [71], for the gas phase reactions, one obtains
4 B_3_H_8_^−^ ➔ B_9_H_14_^−^ + 3 BH_4_^−^ + 3 H_2_
4 B_3_H_8_^−^ ➔ B_10_H_10_^2−^ + 2 BH_4_^−^ + 9 H_2_
exothermic reaction enthalpy values of −413 and −49.8 kJ/mol, respectively, and a strong entropy increase that even further favors the reaction at higher temperatures. These spontaneous overall reaction enthalpies also explain why potential reaction intermediates with 6 to 8 boron atoms are practically not observed. The simultaneous production of BH_4_^−^ in these reactions adds a thermodynamic driving force (as the formation enthalpy of BH_4_^−^ is negative) for these reactions.

In the presence of hydrides, Grinderslev et al. [85] observed the following decomposition reaction at 150 and 200 °C of KB_3_H_8_ under 380 bar of H_2_:KB_3_H_8_ + 2KH ➔ KBH_4_ + K_2_B_12_H_12_ + K_2_B_10_H_10_ + K_2_B_9_H_9_

As shown above, heating solvent-free Mg(B_3_H_8_)_2_ + 4 MgH_2_ either with or without H_2_ gas results in up to 88% back conversion to Mg(BH_4_)_2_ with some MgB_12_H_12_ [60]. These results show that B_3_H_8_^−^ can react in many different ways to either form larger boron hydride clusters or regenerate BH_4_^−^. This can be exploited, for instance, to achieve the direct synthesis of B_10_H_10_^2−^ and B_12_H_12_^2−^ to prepare solid ionic conductors such as Na_4_(B_10_H_10_)(B_12_H_12_), as demonstrated by Gigante et al. [86]. This synthesis starts with the conversion of NaBH_4_ into (Et_4_N)BH_4_, which reacts solvothermally with CH_2_Cl_2_ to form (Et_4_N)B_3_H_8_. (Et_4_N)B_3_H_8_ is then heated in toluene to 185 °C to form a mixture of (Et_4_N)_2_B_10_H_10_ and (Et_4_N)_2_B_12_H_12_, which can then either be separated by fractional crystallization or directly converted with sodium tetraphenylborate into ionic conductor Na_4_(B_10_H_10_)(B_12_H_12_).

## 5. Closoborates and Related Species as Solid Ionic Conductors

Solid ionic conductors for lithium or sodium batteries allow for avoiding the use of a flammable organic electrolyte and are thus expected to considerably improve the safety of batteries. A good solid electrolyte must fulfill several empirical criteria, according to [87]:-“open structure” with a low coordination number of the mobile ion;-The presence of structural phase transitions at low pressure. In the case of AgI, the ambient pressure wurtzite structure (space group P6_3_mc) transforms at 3 kbar and 315 K into a NaCl structure (space group Fm-3m), thus going from a rather covalent network with coordination number 4 to a rather ionic structure with coordination number 6. The associated charge fluctuations between ions can potentially be coupled to vibrational motions and thus dynamically favor ionic conduction.

For practical applications, the conductivity of the material should be higher than 1 mS/cm. Further, the material should have high chemical and thermal stability, and a high electrochemical stability window. Additionally, it must be electronically insulating to avoid battery self-discharge or shortage. Further, the electrolyte should be deformable in order to accommodate the volume changes of anode and cathode materials upon lithium or sodium insertion and removal. This can thus limit the formation of fractures that reduce the performance of the battery. Lastly, the material should not be toxic and be cheap enough for the considered applications.

The discovery of superionic conductivity in the high-temperature phases of LiBH_4_ [88] and Na_2_B_12_H_12_ [29] has stimulated new research for similar compounds with high ionic conductivity at lower temperatures. These compounds include *closo*-hydroborates, *nido*-hydroborates (B_11_H_14_^−^), and *closo*-hydrocarborates (CB_9_H_10_^−^, CB_11_H_12_^−^). Ions B_10_H_10_^2−^ and B_12_H_12_^2−^ are not very toxic. Mutterties et al. [89] reported LD_50_ values for Na_2_B_10_H_10_ and Na_2_B_12_H_12_ administered orally to rats to be around or higher than 7.5 g/kg of body weight for both compounds.

The crystal chemistry of inorganic hydroborates except BH_4_^−^ was recently presented in detail [90], while the crystal chemistry of salts with BH_4_^−^ was addressed in an earlier review [18]: “All nonmolecular hydroborate crystal structures can be derived by simple deformation of the close-packed anionic lattices, i.e., cubic close packing (ccp) and hexagonal close packing (hcp), or bodycentered cubic (bcc), by filling tetrahedral or octahedral sites” [90]. This observation can be illustrated considering group–subgroup relationships of encountered crystal structures, as illustrated in Figure 3 for some relevant compounds [90,91,92,93,94,95,96,97,98,99,100,101,102]. Crystal packing is governed by large anions, leaving in some space groups empty cationic sites, which, of course, favor ionic conduction. For instance, β-Na_2_B_12_H_12_ crystallizes in the Pm-3n space group with a statistical population of 6 sites occupied by 4 Na^+^ ions.

Perturbations of the anionic sublattice further allow for stabilizing the conductive phase at lower temperatures. This was first demonstrated for solid solutions of LiBH_4_ with LiBr and LiI [103]. Phase stability and ionic conductivity in mixed LiBH_4_–LiX (X = Cl, Br) was recently studied in detail [104]. Perturbations of the structure by ball milling or partial substitution was demonstrated for Na_2_B_12_H_12_ with a partial introduction of iodine ions in the *closo*-hydroborate [105]. In a further step, solid solutions of *closo*-hydroborate and *closo*-carbahydroborates, and solid solutions of *nido*-hydroborates with *closo*-hydroborates were studied [106,107,108,109,110,111,112]. Representative examples of mixed borate ionic conductors are shown in Table 1.

The mechanism of ionic conduction in these compounds is related to the dynamical properties of the borohydride or carbohydride ions. These properties can be addressed using NMR [113] and neutron techniques [114], in conjunction with temperature-dependent conductivity and X-ray diffraction, and are supported by theoretical calculations [76,77,88]. A detailed study of ionic conductor Na_4_(B_12_H_12_)(B_10_H_10_) [115] with all these techniques revealed 3 different regimes with increasing temperature. Below −50 °C, conductivity remains very low. Above this temperature, an apparent activation energy of 0.6 eV was found, related to significant couplings of anionic and cationic motions. Above 70 °C, activation energy decreases to 0.37 eV, as thermal energy leads to noncorrelated ionic motions.

One important aspect of solid ionic conductors is their electrochemical stability, which is a critical limit for a reversible battery application. Asakura et al. [116] developed a linear sweep voltammetry method to reliably measure the electrochemical stability of borohydride-based solid electrolytes. The measured oxidative stability of LiBH_4_ of 2.0 V vs. Li^+^/Li was significantly smaller than that in initial reports claiming a stability of up to 5 V [117]. For Na_4_(B_12_H_12_)(B_10_H_10_), two oxidation onsets at 3.02 and 3.22 V vs. Na^+^/Na were tentatively assigned to the onset of decomposition of the less stable [B_10_H_10_]^2−^ and more stable [B_12_H_12_]^2−^ ions, respectively [116]. *Closo*-carborane ions are even more stable, as for Na_4_(CB_11_H_12_)_2_(B_12_H_12_), where a large anodic current was observed above 4 V vs. Na^+^/Na, together with a small onset at 2.93 V. For Li_2_(CB_9_H_10_)(CB_11_H_12_), the onset of decomposition was observed at 2.86 V vs. Li^+^/Li [116]. *Nido*-borates are electrochemically less stable. The oxidative stability limit for Na_5_(B_11_H_14_)(B_12_H_12_)_2_ was 2.6 V vs. Na^+^/Na, and for LiB_11_H_14_, 2.6 V vs. Li^+^/Li [107].

These developments have also led to several all-solid-state battery prototypes based on these mixed borate ionic conductors. Duchêne et al. [118] presented a 3 V sodium battery using Na_4_(B_12_H_12_)(B_10_H_10_), and Murgia et al. [119] showed Na stripping/plating over >500 h in a Na cell with Na_4_(CB_11_H_12_)_2_(B_12_H_12_). Recently, Asakura et al. [120] demonstrated a 4 V sodium battery with the same solid-state conductor, Na_4_(CB_11_H_12_)_2_(B_12_H_12_). These results show that *closo*-hydroborates and their derivatives are very promising materials for chemically and electrochemically stable all-solid-state ionic conductors.

## 6. Conclusions

In the last 20 years, many studies on borohydride species have considerably increased our knowledge on the properties of these materials. For hydrogen storage applications, the kinetics and reversibility of the dehydrogenation reactions remain a major challenge for practical applications. The chemistry of borohydrides from BH_4_^−^ to B_12_H_12_^2−^ in the gas phase and in solution has been theoretically and experimentally addressed; however, in solids, these studies are very challenging, as structural data of potential reaction intermediates such as Mg(B_3_H_8_)_2_ are elusive, and not all intermediates can be observed. If the reaction intermediates are amorphous, X-ray diffraction cannot be used, and theoretical approaches can lead to many different potential structures. The presence of additional hydrides or of Lewis bases such as THF, as shown for the reactions of KB_3_H_8_ and Mg(B_3_H_8_)_2_, strongly modifies the reaction products upon heating. We are thus still very far from a full microscopic understanding of these hydrogenation–dehydrogenation reactions and in the search for optimal catalysts for these processes.

For hydrogen storage, B_3_H_8_^−^ is an interesting species that can be rehydrogenated back to BH_4_^−^. Even though only 25% of the hydrogen is available for this reversible hydrogen storage, the temperatures (less than 200 °C) and kinetics of these reactions approach practical conditions.

The *closo*-hydroborate ions that are formed and identified as intermediates of dehydrogenation reactions have found new and very promising applications as solid-state ionic conductors, as they present many very favorable properties for this use. The recent demonstration of a 4 V all-solid-state battery using solid sodium electrolyte Na_4_(CB_11_H_12_)_2_(B_12_H_12_) [120] highlights this potential. Whether compounds such as Mg(B_10_H_10_), which can be obtained starting from Mg(BH_4_)_2_.2THF, are applicable for new Mg-based batteries remains to be demonstrated. In the preparation of these *closo*-hydroborates and their derivatives, starting from BH_4_^−^ instead of neutral boranes, has the great advantage to reduce the toxicity of the reactants. B_2_H_6_, B_5_H_9_ and B_10_H_14_ are highly toxic and thereby not really suitable for industrial production processes of *closo*-hydroborates at a higher scale. Thus, boron–hydrogen compounds have a future for new green energy applications.

## Figures and Tables

**Figure 1 molecules-26-07425-f001:**
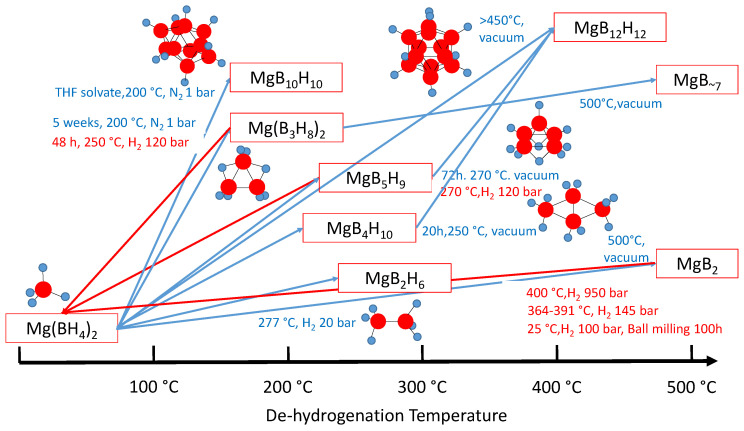
Illustration of Mg(BH_4_)_2_ dehydrogenation reactions (blue arrows) and rehydrogenation reactions (red arrows) reported in the literature [22,34,35,36,37,38,39,40,41,42,43,44]. Upon further heating, these intermediate species, which are associated with (amorphous) MgH_2_, form MgB_2_.

**Figure 2 molecules-26-07425-f002:**
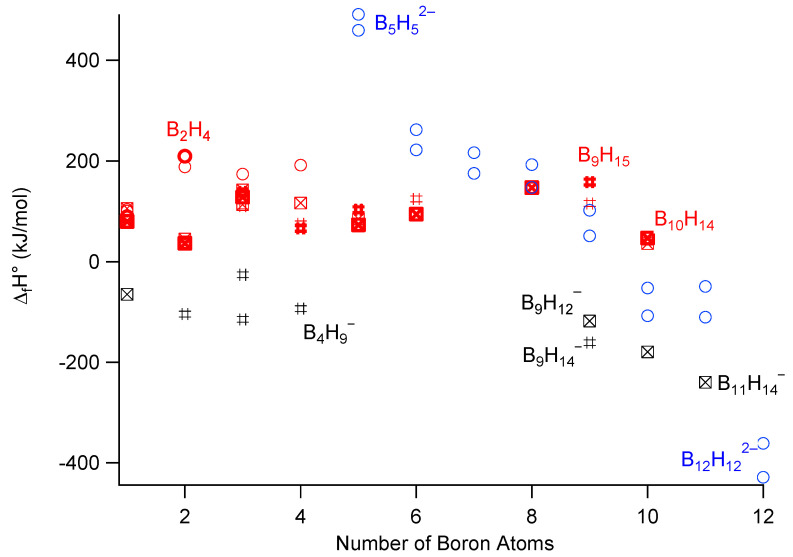
Experimental (bold) and theoretical formation enthalpy values for neutral (red) monoanionic (black) and dianionic (blue) species. *Closo* species, circles; *nido,* #; *arachno,* crossed squares. Data from [69,70,71,72,74,75,76]. For *closo* ions B_n_H_n_^2−^, data (blue circles) from 2 different studies [69,72] reveal systematic differences. All monoanionic species (in black) have negative formation enthalpies, while all neutral boranes (in red) have positive formation enthalpy.

**Figure 3 molecules-26-07425-f003:**
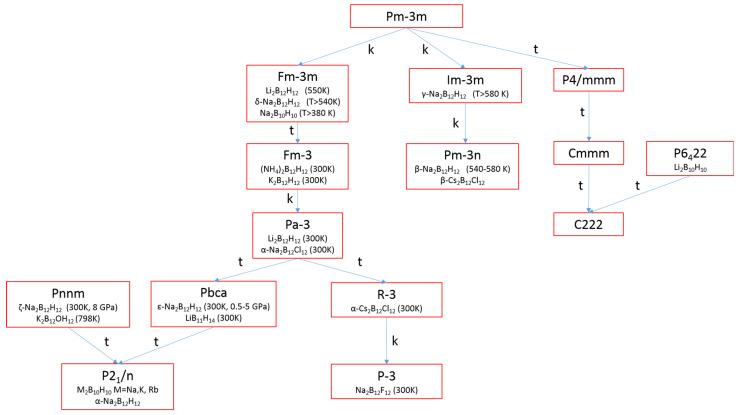
Group–subgroup relationships between space groups (in Herrmann-Mauguin notation) of *closo*-hydro borates and some *closo*-halogeno borates. t, “translationengleich” subgroups; k, “klassengleich” subgroups.

**Table 1 molecules-26-07425-t001:** Examples of ionic conductivity in mixed borate salts.

Compound	Temperature	Conductivity	Reference
0.7 Li(CB_9_H_10_)–0.3 Li(CB_11_H_12_)	298 K	6.7 mS/cm	[106]
Li_2_(B_11_H_14_)(CB_11_H_12_)	298 K	0.11 mS/cm	[107]
Li_3_(B_11_H_14_)(CB_11_H_12_)_2_	298 K	1.1 mS/cm	[107]
Na_3_(CB_11_H_12_)(B_12_H_12_)	298 K	2 mS/cm	[108]
Na_4_(CB_11_H_12_)_2_(B_12_H_12_)	298 K	2 mS/cm	[108]
Na_4_(B_10_H_10_)(B_12_H_12_)	298 K	0.9 mS/cm	[109]
Na_2_(B_10_H_10_)−3 Na_2_(B_12_H_12_)	298 K	0.34 mS/cm	[110]
Na_x+2y_(B_11_H_14_)_x_(B_12_H_12_)_y_	298 K	3–4 mS/cm	[111]

## Data Availability

Not applicable.

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
