# Peer review of "Boron Hydrogen Compounds: Hydrogen Storage and Battery Applications"

_molecules, 2021, doi:10.3390/molecules26247425_

Round 1

Reviewer 1 Report

The present manuscript is a mini-review referring some of the aspects of chemistry of boron hydrogen compounds. In particular - magnesium borohydride and its derivatives, the intermediate products of thermal decomposition of borohydride anion, and closoborates and related species as solid ion conductors. While the review remains not comprehensive for these very broad topics, it could still be useful for the community.

Therefore, I would recommend its publication after some minor improvements listed below.

  1. The abstract should be more informative. It should be rewritten to better reflect the content of the manuscript.
  2. Some of the de-hydrogenation products presented in Figure 1 deserve an explanation, for example MgB~7.
  3. While the THF and related solvates rather destabilize Mg(BH4)2, its derivatives containing organic cations are rather more stable. This feature is important both from the point of view of hydrogen storage and battery applications and has been reported for several systems, cf.: doi.org/10.1039/C6DT02239A and doi.org/10.1021/acsaem.0c02861. The author should include these findings in the discussion.
  4. The manuscript should be corrected for the spelling, minor language and referencing mistakes, for example, p. 7 line 267 - ref. [73] is not a review, and should be replaced by the other one.

Author Response

We thank the reviewers for their constructive suggestions. Please find below our corrections done.

Reviewer 1:

The present manuscript is a mini-review referring some of the aspects of chemistry of boron hydrogen compounds. In particular - magnesium borohydride and its derivatives, the intermediate products of thermal decomposition of borohydride anion, and closoborates and related species as solid ion conductors. While the review remains not comprehensive for these very broad topics, it could still be useful for the community.

Therefore, I would recommend its publication after some minor improvements listed below.

The abstract should be more informative. It should be rewritten to better reflect the content of the manuscript.

The abstract has been corrected as follows:

About 25 years ago, Bogdanovic and Schwickardi (B. Bogdanovic, M. Schwickardi: J. Alloys Compd. 1–9, 253 (1997) discovered the catalyzed release of hydrogen from NaAlH4. This discovery stimulated a vast research effort on light hydrides as hydrogen storage materials, in particular boron hydrogen compounds. Mg(BH4)2 with a hydrogen content of 14.9 wt% has been extensively studied, and recent results shed new insights on intermediate species formed during dehydrogenation. The chemistry of B3H8- which is an important intermediate between BH4- and B12H122- is presented in detail. The discovery of high ionic conductivity in the high temperature phases of LiBH4 and Na2B12H12 opened a new research direction. The high chemical and electrochemical stability of closo-hydroborates has stimulated new research for their applications in batteries and very recently a all-solid-state 4V Na battery prototype using a Na4(CB11H12)2(B12H12) solid electrolyte has been demonstrated. In this review, we present the current knowledge of possible reaction pathways involved in the successive hydrogen release reactions from BH4- to B12H122-, as well as a discussion of relevant properties necessary for high ionic conduction materials.

Some of the de-hydrogenation products presented in Figure 1 deserve an explanation, for example MgB~7.

Boron-rich MgB7 films are obtained by heating volatile Mg(B3H8)2 solvates with dimethylether and diethylether. (Girolami Inorg Chem 2007). This is now mentioned in the text.

While the THF and related solvates rather destabilize Mg(BH4)2, its derivatives containing organic cations are rather more stable. This feature is important both from the point of view of hydrogen storage and battery applications and has been reported for several systems, cf.: doi.org/10.1039/C6DT02239A and doi.org/10.1021/acsaem.0c02861. The author should include these findings in the discussion.

This was done and the corresponding references were added.

Mixtures of Mg(BH4)2 with (CH3)4NBH4 (5:1 molar) reveal reversible melting around 180-195 °C [Bell et al] with enhanced stability compared to melts of pure Mg(BH4)2 and (CH3)4NBH4.  [Ph4P]2[Mg(BH4)4] gradually looses mass over 225-230 °C, but heating to 500 °C does not lead to the mass loss expected for the formation of MgB2. A similar behavior was observed for [Me4N]2[Mg(BH4)4]. [Grochala] These findings suggest that derivates of Mg(BH4)2 with organic cations are rather stabilized.

The manuscript should be corrected for the spelling, minor language and referencing mistakes, for example, p. 7 line 267 - ref. [73] is not a review, and should be replaced by the other one.

Thank you for pointing out this error. The references were rechecked.

Reviewer 2 Report

This review paper deals with the application of boron hydrogen compounds as potential materials for hydrogen storage and as solid ionic conductors. This topic is of great importance in modern inorganic chemistry. Boron hydrogen compounds are extensively studied both theoretically and experimentally (10.1021/jacs.0c06159, 10.1002/anie.201915290, 10.3390/cancers12113423). There are a lot of reviews analyzing the structure and chemical behavior of these compounds as well as their potential application in medicine, energetics etc (10.1016/j.ccr.2021.214042, 10.3390/molecules25040828, 10.1039/c9cs00197b). In addition, excellent reviews devoted to hydrogen storage and solid ionic conductors have been published recently (10.1016/j.ijhydene.2019.01.104, 10.1021/acs.chemrev.8b00313, 10.1088/2516-1083/ab73dd). The key difference of the present paper from those previously published is much more attention to the application of higher closo-borate clusters such as [B10H10]2– and [B12H12]2–

Previously, several works dealing with the chemistry of closo-borates anions and related compounds have been published in «Molecules» (10.3390/molecules26123754, 10.3390/molecules26010248, 10.3390/molecules25246009). Current review continues and expands investigations in the field of boron hydrogen compounds. Presented generalization of the progress in the application of boron hydrides is useful for the chemists focused in hydride chemistry.

Despite of obvious advantages of manuscript some important points should be improved:

  1. The title does not seem correct. «… from hydrogen storage to battery applications» implies a description of a wide range of applications. Whereas the paper lists only two areas of potential applications. Perhaps a somewhat better title would be «Boron hydrogen compounds: hydrogen storage and battery applications».
  2. In the introductory part, it is necessary to indicate clearly the motivation for this work. Why did the author choose these applications to describe in the paper? In general, hydrogen storage materials and solid ionic conductors are rather different types of application. These are two different fields with their own methods and approaches. This circumstance is also well reflected in the review. A brief part should be added to the work for logically combining two unbound sections.
  3. Much attention is paid to magnesium borohydride. It is worth pointing out why other borohydride-based salts are not mentioned.
  4. Figure 1 seems a bit sloppy. It is quite asymmetrical, its parts are arranged chaotically. All the arrows lead to magnesium borohydride, but the compounds listed can also turn into each other. This is worth pointing out, e.g. by adding another diagram. It is necessary to add graphical representations of the structures of compounds. Just, for example, the structure of MgB2H6 may not be familiar to a wide range of readers.
  5. All terms such as closo- and nido- should be italicized.
  6. Figure 2 is a little bit difficult to grasp. Everything overlaps on top of each other. The author needs to add a legend for better reader understanding.
  7. The conclusions need to be changed. It is necessary to state which of the boron hydrogen compounds are the most promising for the production of energy materials. In addition, the author can briefly point out the main advantages of the most perspective compounds.

To conclude, this paper is interesting and useful and deserves to be published in «Molecules» after the minor revision.

Author Response

Reviewer 2.

The references indicated by the reviewer were included.

Despite of obvious advantages of manuscript some important points should be improved:

The title does not seem correct. «… from hydrogen storage to battery applications» implies a description of a wide range of applications. Whereas the paper lists only two areas of potential applications. Perhaps a somewhat better title would be «Boron hydrogen compounds: hydrogen storage and battery applications».

The title was changed as suggested.

In the introductory part, it is necessary to indicate clearly the motivation for this work. Why did the author choose these applications to describe in the paper? In general, hydrogen storage materials and solid ionic conductors are rather different types of application. These are two different fields with their own methods and approaches. This circumstance is also well reflected in the review. A brief part should be added to the work for logically combining two unbound sections.

This was done as follows:

The dehydrogenation reactions of metal borohydrides ultimately lead to hydrogen, metal and boron or metal borides. In this reaction process, intermediate species are formed, in particular compounds with the closo-hydroborate anion B12H122- [Wolverton ref.17, Hwang, S.-J.; Bowman, R. C.; Reiter, J. W.; Rijssenbeek, J.; Soloveichik, G. L.; Zhao, J.-C.; Kabbour, H.; Ahn, C. C. NMR confirmation for formation of [B12H12]2‑ complexes during hydrogen desorption from metal borohydrides. J. Phys. Chem. C 2008, 112,3164−3169]. B12H122- is particularly stable, and can therefore also act as a detrimental thermodynamic sink for further dehydrogenation reactions. This fact stimulated new research on the thermal properties of closo-hydroborate salts which revealed a high temperature phase transition in Na2B12H12 leading to a superionic phase [ref]. Thus, the controlled dehydrogenation of a borohydride salt can be used to prepare safely new closo- and nido- hydroborate salts for potential battery applications without using toxic boranes such as B10H14 which have previously been used for the synthesis of this large boron species [He, L.; Li, H.-W.; Hwang, S.-J.; Akiba, E. Facile Solvent-Free Synthesis of anhydrous alkali metal dodecaborate M2B12H12 (M= Li,Na, K). J. Phys. Chem. C 2014, 118, 6084−6089]

Much attention is paid to magnesium borohydride. It is worth pointing out why other borohydride-based salts are not mentioned.

Hydrogen storage in other borohydrides such as LiBH4 has been reviewed recently [Zhang, Wenxuan; Zhang, Xin; Huang, Zhenguo; Li, Hai-Wen; Gao, Mingxia; Pan, Hongge; Liu Yongfeng, Recent Development of Lithium Borohydride-Based Materials for Hydrogen Storage, Adv. Energy Sustainability Res. 2021, 2, 2100073]. Recent results  on potential dehydrogenation intermediates have been reported for Mg(BH4)2 and provide new insights on potential reaction intermediates and are reported herein. 

Figure 1 seems a bit sloppy. It is quite asymmetrical, its parts are arranged chaotically. All the arrows lead to magnesium borohydride, but the compounds listed can also turn into each other. This is worth pointing out, e.g. by adding another diagram. It is necessary to add graphical representations of the structures of compounds. Just, for example, the structure of MgB2H6 may not be familiar to a wide range of readers.

Figure 1 has been redrawn with the structures of the borohydride ions

All terms such as closo- and nido- should be italicized.

This has been done

Figure 2 is a little bit difficult to grasp. Everything overlaps on top of each other. The author needs to add a legend for better reader understanding.

This has been done.

The conclusions need to be changed. It is necessary to state which of the boron hydrogen compounds are the most promising for the production of energy materials. In addition, the author can briefly point out the main advantages of the most perspective compounds.

This has been done by adding the following text in the conclusions:

For hydrogen storage, B3H8- appears to be an interesting species which can be rehydrogenated back to BH4-. Even though only 25% of the hydrogen is available for this reversible hydrogen storage, the temperatures (less than 200 °C) and kinetics of these reactions approach practical conditions.

The recent demonstration of a 4V all solid state battery using the solid sodium electrolyte Na4(CB11H12)2(B12H12) [121] highlights this potential. Whether compounds such as Mg(B10H10), which can be obtained starting from Mg(BH4)2.2THF, will be applicable for new Mg-based batteries remains to be demonstrated.